# Pramipexole Hyperactivates the External Globus Pallidus and Impairs Decision-Making in a Mouse Model of Parkinson’s Disease

**DOI:** 10.3390/ijms25168849

**Published:** 2024-08-14

**Authors:** Hisayoshi Kubota, Xinzhu Zhou, Xinjian Zhang, Hirohisa Watanabe, Taku Nagai

**Affiliations:** 1Division of Behavioral Neuropharmacology, International Center for Brain Science (ICBS), Fujita Health University, Toyoake 470-1192, Aichi, Japan; hisayoshi.kubota@fujita-hu.ac.jp (H.K.);; 2Department of Neurology, School of Medicine, Fujita Health University, Toyoake 470-1192, Aichi, Japan

**Keywords:** c-Fos, decision-making, dopamine D3 receptor, DREADD, impulse control disorders, Iowa Gambling Task, Parkinson’s disease, pathological gambling, pramipexole, 6-hydroxydopamine

## Abstract

In patients with Parkinson’s disease (PD), dopamine replacement therapy with dopamine D2/D3 receptor agonists induces impairments in decision-making, including pathological gambling. The neurobiological mechanisms underlying these adverse effects remain elusive. Here, in a mouse model of PD, we investigated the effects of the dopamine D3 receptor (D3R)-preferring agonist pramipexole (PPX) on decision-making. PD model mice were generated using a bilateral injection of the toxin 6-hydroxydopamine into the dorsolateral striatum. Subsequent treatment with PPX increased disadvantageous choices characterized by a high-risk/high-reward in the touchscreen-based Iowa Gambling Task. This effect was blocked by treatment with the selective D3R antagonist PG-01037. In model mice treated with PPX, the number of c-Fos-positive cells was increased in the external globus pallidus (GPe), indicating dysregulation of the indirect pathway in the corticothalamic-basal ganglia circuitry. In accordance, chemogenetic inhibition of the GPe restored normal c-Fos activation and rescued PPX-induced disadvantageous choices. These findings demonstrate that the hyperactivation of GPe neurons in the indirect pathway impairs decision-making in PD model mice. The results provide a candidate mechanism and therapeutic target for pathological gambling observed during D2/D3 receptor pharmacotherapy in PD patients.

## 1. Introduction

Parkinson’s disease (PD) is a neurodegenerative disorder of dopaminergic neurons in the nigrostriatal pathway, characterized by motor symptoms including rigidity, tremors, and bradykinesia [1]. Dopamine replacement therapy with dopamine D2/D3 receptor agonists is commonly used to alleviate these symptoms, but side effects include the development of impulse control disorders (ICDs) in a significant number of patients [2,3]. ICDs are defined as the persistent inability to resist the urge to engage in behaviors that lead to harmful personal, social, or financial outcomes, and comprise pathological gambling, hypersexuality, binge eating, and compulsive shopping [3]. Indeed, the prevalence of pathological gambling among patients with PD is considerably higher than that of the general population [4,5].

Pathological gambling is thought to be rooted in an impairment of decision-making [6], a multifaceted cognitive process involving the selection of one choice among several alternatives, often with a risk/reward tradeoff. In humans, decision-making can be assessed using the Iowa Gambling Task (IGT), a behavioral assay that mimics real-life situations by reproducing uncertain conditions based on probabilistic rewards or penalties. In the IGT, the optimal strategy is to favor options with smaller gains and penalties while avoiding “high-risk, high-reward” options. However, PD patients under pharmacotherapy show a preference for risky choices, suggesting a decision-making impairment [7,8]. In general, the neural circuitry underlying decision-making in the rodent IGT overlaps with that in humans [9,10,11,12], indicating that the IGT has translational validity for exploring circuit mechanisms in rodent PD models.

Corticothalamic-basal ganglia circuits play a central role in decision-making processes [13,14]. Medium spiny neurons (MSNs) are the predominant GABAergic inhibitory cell, accounting for 95% of neurons in the striatum (STR). MSNs are further divided into dopamine D1 receptor-expressing MSNs (D1R-MSNs) and dopamine D2 receptor-expressing MSNs (D2R-MSNs). Cortical activation influences MSNs, which then trigger outputs via the direct and indirect pathways that provide cortical feedback output via the thalamus (Th). In the direct pathway, D1R-MSNs project to the internal globus pallidus (GPi) and substantia nigra pars reticulata (SNr), while in the indirect pathway, D2R-MSNs project to the external globus pallidus (GPe). These two pathways exert opposing regulatory effects on decision-making by modulating the firing rate of basal ganglia output nuclei. Both types of MSNs express the dopamine D3 receptor (D3R), which inhibits intracellular signaling via the G-protein Gi [15]. In preclinical studies, dopamine D2/D3 receptor agonists pramipexole (PPX) and ropinirole induce risk-taking behaviors in rodents [16,17]. Treatment of PD patients with PPX, which has a higher affinity for D3R than ropinirole [18], is strongly associated with an increased risk of ICDs [3]. Moreover, a rat IGT study indicates that a D3R agonist increases disadvantageous choices, while a D3R antagonist has the opposite effect [19]. Together, these findings suggest that D3R signaling plays an important role in decision-making impairments. However, the neural circuit mechanisms by which treatment with PPX impairs decision-making within the corticothalamic-basal ganglia circuitry remain unclear.

The present study employed a PD mouse model to investigate the effects of PPX on the corticothalamic-basal ganglia circuitry underlying decision-making in a touchscreen-based IGT. Our findings implicate hyperactivation of the GPe in the indirect pathway as a candidate neural mechanism by which D3R agonists like PPX can lead to decision-making impairments.

## 2. Results

### 2.1. Effect of PPX Treatment on Decision-Making in Sham- and 6-OHDA-Lesioned Mice

Touchscreen-based behavioral testing has become increasingly popular as an automated, translatable, and reproducible method for assessing cognition in animal models of neuropsychiatric disease [20]. Here, we employed a touchscreen-based IGT to assess decision-making in a partial nigrostriatal lesion model of PD. Mice were trained to develop screen-touching behaviors and to become familiar with the reward/punishment probabilities associated with each choice in the IGT (Figure 1). Two panel choices (P1 and P2) resulted in small rewards and low punishments, leading to an advantageous outcome, while the other two panel choices (P3 and P4) resulted in high rewards and high punishments, leading to a disadvantageous outcome. After meeting the criteria for IGT training, 6-hydroxydopamine (6-OHDA) (4 μg/site) was microinjected bilaterally into the dorsolateral STR of the mice (Figure 2A,B). We confirmed that the injection of 6-OHDA significantly decreased the number of tyrosine hydroxylase (TH)-positive cells in the STR (Figure 2C,D) compared with those of the sham-lesioned mice. Fourteen days after the microinjection, sham- or 6-OHDA-lesioned mice were administered PPX (0.3 mg/kg) for 10 consecutive days. Mouse behavior, including decision-making, motor impulsivity, attention, and motivation were evaluated using the IGT during two distinct periods: the first 5 days (Period 1) and the subsequent 5 days (Period 2) (Figure 2E–M, Table 1). There was no significant difference between sham/SAL and 6-OHDA/SAL in the percentage of disadvantageous choices during Period 1 and Period 2, although 6-OHDA/SAL mice tended to increase the percentage of disadvantageous choices (P3 + P4) (Figure 2E). Sham-lesioned mice treated with PPX exhibited impaired decision-making, as evidenced by an increase in disadvantageous choices (P3 + P4) and premature responses during Period 2, but not Period 1 (Figure 2E,H,K). Surprisingly, PPX treatment also impaired decision-making in 6-OHDA-lesioned mice, as evidenced by an increase in response to P4, the highest risky choice, in both Period 1 and Period 2 (Figure 2E,I). Although PPX decreased responses in P2 and total trial initiations and increased omissions in 6-OHDA-lesioned mice in Period 1 or Period 2, there was no difference in the latency to collect rewards (Figure 2G,J,L,M). Overall, PPX treatment impaired decision-making in control and test mice, but was strongly enhanced in the 6-OHDA-lesioned mice.

### 2.2. Effect of D3R Antagonist on PPX-Induced Decision-Making Impairments in 6-OHDA-Lesioned Mice

We investigated the effect of different doses of PPX (0.01, 0.1, or 0.3 mg/kg) on disadvantageous choice behavior to understand the dose–response relationship of PPX-induced impairments in decision-making (Figure 3A). PPX treatment at a dose of 0.3 mg/kg increased disadvantageous choices, while doses of 0.01 and 0.1 mg/kg of PPX had no effect on disadvantageous choices in sham- and 6-OHDA-lesioned mice (Figure 3B).

Considering that PPX is a high-affinity dopamine D3R agonist [18], we next investigated whether PPX treatment impairs decision-making in 6-OHDA-lesioned mice through D3R by co-administering a selective D3R antagonist PG-01037 (Figure 3C). The results show that PG-01037 suppressed the increase in disadvantageous choices induced by PPX treatment (Figure 3D). These findings suggest that PPX treatment at a dose of 0.3 mg/kg impairs decision-making in 6-OHDA-lesioned mice selectively through D3R.

### 2.3. Expression of D3R in the STR of PPX-Treated Sham- and 6-OHDA-Lesioned Mice

It is possible that chronic PPX treatment or the 6-OHDA lesion could alter the levels of D3R in the brain [21,22]. To examine this possibility, we used quantitative PCR to determine the levels of D3R mRNA in the STR of 6-OHDA-lesioned mice with repeated PPX treatment. PPX treatment decreased the levels of D3R mRNA in the STR of both sham- and 6-OHDA-lesioned mice (Appendix A). However, the 6-OHDA lesion itself had no effect on the levels of D3R mRNA among the groups (Appendix A).

### 2.4. c-Fos Mapping in the Brain of PPX-Treated Sham- and 6-OHDA-Lesioned Mice

We mapped the brain regions associated with PPX-induced impairments of decision-making by measuring c-Fos-positive cells in the brains of PPX-treated sham- and 6-OHDA-lesioned mice. Brain samples were collected 2 h after performance of the IGT on the last day of Period 2 (Figure 4, Appendix A). Our findings indicate an overall decrease in the number of c-Fos-positive cells in the STR of both sham- and 6-OHDA-lesioned mice treated with PPX (Figure 4A,B). However, an increase in the number of c-Fos-positive cells was observed in the GPe of both sham- and 6-OHDA-lesioned mice treated with PPX (Figure 4C,D). Significant main effects of PPX on the number of c-Fos-positive cells were observed in several brain regions, including in the intermediate part of the lateral septum (LS), CA1-3, and dentate gyrus subregions of the hippocampus (HIP) and Th, in both sham- and 6-OHDA-lesioned mice treated with PPX (Appendix A). The changes in c-Fos-positive cells induced by PPX treatment between sham- and 6-OHDA-lesioned mice were comparable (Figure 4, Appendix A).

### 2.5. Effect of Chemogenetic Inhibition of GPe on PPX-Induced Decision-Making Impairments in 6-OHDA-Lesioned Mice

Based on the c-Fos mapping results, we hypothesized that neural circuit dysfunctions in the STR-GPe or HIP-LS pathway may be associated with the PPX-induced decision-making impairments in 6-OHDA-lesioned mice [13,23,24,25]. In order to manipulate these brain areas, we used a κ-opioid receptor-based DREADD (KORD) coupled to the G-protein Gi, which can be activated by the specific ligand salvinorin B (SALB) [26]. Prior to the IGT training, we bilaterally injected AAV-SYN1-KORD-P2A-mCherry into the GPe or LS of mice (Figure 5A,B and Appendix A). We confirmed that SALB administration significantly reduced the percentage of double-positive cells for both mCherry and c-Fos in the GPe of 6-OHDA-lesioned mice treated with PPX, demonstrating the inhibitory effect of KORD on neuronal activity (Figure 5C–E). We next investigated the effect of KORD-mediated inhibition of the GPe or LS on PPX-impaired decision-making in 6-OHDA-lesioned mice. Treatment with SALB in 6-OHDA-lesioned mice expressing KORD in the GPe, but not in the LS, suppressed PPX-induced potentiation of disadvantageous choices (Figure 5F and Appendix A). These findings strongly suggest that hyperactivation of the GPe is responsible for PPX-induced impairment of decision-making in 6-OHDA-lesioned mice.

## 3. Discussion

Dopamine replacement therapy with dopamine D2/D3 receptor agonists is a common treatment for PD. However, in clinical settings, these agonists can induce pathological gambling associated with decision-making impairments [3,7,8]. The present study examined the neural mechanisms underlying D3R-based decision-making impairments with the D3R-preferring agonist PPX in 6-OHDA-lesioned mice. PPX induced disadvantageous choices in the touchscreen-based IGT that were prevented by an D3R antagonist. c-Fos mapping indicated that PPX resulted in hyperactivation of the GPe, while DREADD-based inhibition of the GPe rescued both c-fos staining and disadvantageous choice behavior. These findings provide a working model of D3R action on the STR, where PPX impairs decision-making by disinhibiting GABAergic D2R-MSNs in the indirect pathway of the STR via the D3R and causes hyperactivation of the GPe (Figure 6). The current study represents a significant step toward the development of clinical strategies for maintaining the therapeutic efficacy of PPX and other D2/D3R agonists for PD while reducing their adverse effects on ICDs.

Our study reveals new insights into the neural circuit mechanisms of the GPe within the STR and basal ganglia loops. The corticothalamic-basal ganglia circuitry is primarily divided into motor, associative, and limbic loops [27,28], where the GPe serves as a relay nucleus, receiving GABAergic inputs from striatal D2R-MSNs in the indirect pathway, which express the Gi-coupled D3R [15]. Evidence from primate studies indicates that GPe subregions have specific roles in movement, action selection, and motivation via connections with distinct regions of the cortex and Th [29]. In this study, the suppression of neuronal activity specifically in the STR suggests that PPX acts at D3Rs on striatal D2R-MSNs, leading to GABAergic disinhibition within the STR and hyperactivation of the GPe (Figure 4 and Figure 6) in the indirect pathway. We found that KORD-mediated inhibition of the GPe ameliorated PPX-induced disadvantageous choices in 6-OHDA-lesioned mice (Figure 5F), suggesting that GPe hyperactivation impairs decision-making by interfering with the learning of negative outcomes, as has been observed in PD patients treated with PPX [30,31].

Behavioral addictions induced by dopamine agonism can be attributed to other major dopamine sources, including the mesolimbic system from the ventral tegmental area (VTA) to the nucleus accumbens (NAc), which is not specifically degenerated in PD. In this system, the NAc D1R-MSN is responsible for regulating reward, motivation, and positive reinforcement [32,33]. Our c-Fos mapping did not reveal any alteration in the VTA and NAc of PPX-treated mice (Appendix A). Accordingly, the mechanism by which PPX induced decision-making impairments may differ from that of behavioral addictions. However, a different interpretation may be provided with real-time observation of individual neuronal activity in the mesolimbic system during the IGT using in vivo recording (e.g., calcium imaging). This should be investigated in future studies.

While our findings highlight the role of the GPe in decision-making [34], we also observed increased activity in the HIP-LS circuit in PPX-treated mice (Appendix A). However, pharmacogenetic inhibition of the LS activity by KORD failed to ameliorate decision-making impairments (Appendix A). Considering that the HIP-LS circuit is known to modulate social and depressive behaviors [35,36,37], we hypothesize that activation of the LS may be responsible for mechanisms other than decision-making, such as the antidepressant effects of PPX [38,39], and this should be further examined in future studies.

In humans, PPX treatment has been reported to promote impulsivity and pathological gambling, independent of any underlying PD pathology [40,41]. Consistent with these previous reports [16,42], we showed that PPX treatment induced motor impulsivity and disadvantageous risky choices in sham-lesioned control mice (Figure 2E,H,K). Notably, the adverse effects of PPX were potentiated in 6-OHDA-lesioned mice (Figure 2E,I). However, the potentiating effect of PPX was suppressed by co-administration with the selective D3R antagonist PG-01037 (Figure 3D). Impairment of the higher cortical regions involved in decision-making has been demonstrated in rodent models of PD [43,44,45]. This may support, at least in part, our working model (Figure 6) and contribute to the potentiating effect of PPX in 6-OHDA-lesioned mice. Another hypothesis is that D3R expression and activity could be altered in 6-OHDA-lesioned mice. Prior studies demonstrated that D3R expression is increased in the brains of PD model rats treated with PPX [21,22]; however, we showed that PPX decreased the expression levels of D3R mRNA in both sham- and 6-OHDA-lesioned mice (Appendix A). This discrepancy might be attributed to differences in animal species, brain regions analyzed, and experimental protocols, such as the dose of PPX applied and the duration of treatment [21,22]. It is also possible that dopamine depletion may enhance D3R signaling by altering the membrane trafficking system [46,47,48] or the interaction of D3R with the splice variant D3nf, which can reduce its ligand-binding capacity [49,50]. Based on these considerations, further investigation of the relationship between D3R signaling and PPX-induced behavioral abnormalities should clarify these questions. Since humans and rodents exhibit comparable variability in individual choice preferences in the IGT [51], our touchscreen-based IGT may help to evaluate such mechanisms of decision-making.

The PD model mice we generated were translationally compatible with the IGT. Unilateral injection of 6-OHDA into the medial forebrain bundle, which results in total denervation of the dopaminergic nigrostriatal pathway, has been extensively used to study motor deficits associated with parkinsonism [52,53]. We used a partial nigrostriatal lesion model of PD (Figure 2B–D) to analyze non-motor behaviors [39,54] and avoid disrupting the experimental conditions of the IGT by eliminating the effects of distorted posture and unilateral movement observed in unilateral lesions. However, it is important to note that this model exhibits a depression-like phenotype [39] which may impact the affective context in the IGT. Although 6-OHDA-lesioned mice showed a reduction in total trial initiation and an increase in omission frequency (Figure 2J,L), the consistent collection latency in 6-OHDA-lesioned mice suggests no discernible influence on reward reinforcers or motivation to perform the IGT (Figure 2M). PPX treatment increased disadvantageous risky choices in 6-OHDA-lesioned mice (Figure 2E,I), which is consistent with clinical observations that patients with PD who had never gambled before develop pathological gambling after initiating dopamine agonist therapy [55].

In conclusion, our study demonstrates the importance of the corticothalamic-basal ganglia circuitry in regulating decision-making in healthy and PD subjects. Our findings indicate that PPX targets D3Rs on striatal MSNs, causing GABAergic disinhibition within the STR, resulting in hyperactivation of the GPe in the indirect pathway, which may impair decision-making and cause ICDs. Overall, our findings pinpoint a neural circuit involved in decision-making and a candidate therapeutic target for PPX-induced pathological gambling.

## 4. Materials and Methods

### 4.1. Mice

Male mice were used to exclude any potential estrous cycle effects in female mice [56]. C57BL/6N mice were obtained from Japan SLC (Shizuoka, Japan). The mice began the IGT training at 8 weeks of age and were 16–18 weeks of age at the time of testing. The mice were housed in a specific pathogen-free environment within our animal facility prior to use and were maintained in a regulated environment (23 ± 3 °C and 50 ± 10% humidity) with a 12 h light/dark cycle (lights on at 8:00 a.m., off at 8:00 p.m.). The mice were provided food (MF; Oriental Yeast Co., Ltd., Tokyo, Japan) and tap water ad libitum. All animal experiments were approved (number: AP20002-MD3) and performed in accordance with the guidelines for the care and use of laboratory animals established by the Animal Experiments Committee of Fujita Health University.

### 4.2. Drug Administration

PPX (Tokyo Chemical Industry, Tokyo, Japan, #P2073) was dissolved in 0.9% saline. PG-01037 (Cayman Chemicals, Ann Arbor, MI, USA, #22079) was dissolved in 5% dimethyl sulfoxide (DMSO). PPX [0.01, 0.1, or 0.3 mg/kg, subcutaneous (s.c.)] and PG-01037 [10 mg/kg, intraperitoneal (i.p.)] were administered systemically 60 and 90 min before the behavioral test, respectively. Drug dosages and experimental schedules were determined based on previous reports [38,39,57].

### 4.3. 6-OHDA Lesion

6-OHDA lesions were performed in accordance with previous reports [39,54]. 6-OHDA hydrochloride (4 μg/μL, Santa Cruz Biotechnology, Santa Cruz, CA, USA, #SC-203482) was dissolved in 0.9% saline containing 0.02% ascorbic acid. In brief, mice were pretreated with desipramine (25 mg/kg, i.p., Millipore, Billerica, MA, USA, #D3900) 30 min prior to the injection of 6-OHDA to protect norepinephrine nerve terminals. The mice were anesthetized by i.p. administration of a mixture of anesthetic agents containing medetomidine (0.75 mg/kg, Nippon Zenyaku Kogyo, Fukushima, Japan), midazolam (4 mg/kg, Sandoz, Basel, Switzerland), and butorphanol (5 mg/kg, Meiji Seika Pharma, Tokyo, Japan), and positioned in a stereotaxic frame (David Kopf, Tujunga, CA, USA). 6-OHDA was microinjected bilaterally into the dorsolateral STR (+0.6 mm anterior-posterior, ±2.2 mm medial-lateral from the bregma, and −3.2 mm dorsal-ventral from the skull) in a volume of 1 μL/site, according to the mouse brain atlas. The control sham-lesioned mice were injected with the same volume of vehicle. Two weeks after the injections, the mice were subjected to the IGT.

### 4.4. Adeno-Associated Virus Preparation

pAAV[Exp]-SYN1>HA/[KORD(ns)]:P2A:mCherry:WPRE was constructed and packaged by VectorBuilder (Kanagawa, Japan). The KORD cDNA sequence, followed by the self-cleaving 2A peptide sequence and mCherry sequence, were cloned downstream of the SYN1 promotor in pAAV[Exp]-SYN1:WPRE. Plasmids for the AAV vector, pHelper, and pAAV-8 were transfected into AAV293 cells. After incubation, cells were collected and purified by performing CsCl-gradient ultracentrifugation. CsCl was removed via dialysis. AAV titers were estimated via a quantitative polymerase chain reaction. The viral vector was subdivided into aliquots and stored at −80 °C until use.

### 4.5. In Vivo Chemogenetic Manipulation

The protocol was previously described in [26]. Briefly, mice were anesthetized with a mixture of anesthetic agents and positioned in a stereotaxic frame. AAV-SYN1-KORD-P2A-mCherry (1 × 10^12^ genome copies/mL) was microinjected bilaterally into the GPe (+0.6 mm anterior-posterior, ±2.2 mm medial-lateral from the bregma, and −3.2 mm dorsal-ventral from the skull) or LS (+0.6 mm anterior-posterior, ±2.2 mm medial-lateral from the bregma, and −3.2 mm dorsal-ventral from the skull) in a volume of 0.5 μL/site. Behavioral experiments were initiated at least 2 weeks after the virus injection in order to ensure stable transgene expression. Test mice were administered the vehicle (5% DMSO in 0.9% saline, s.c.) or SALB (10 mg/kg, s.c., Cayman Chemicals, #23582) 10 min prior to the IGT.

### 4.6. Touchscreen-Based IGT

#### 4.6.1. Apparatus

The mouse Iowa Gambling Task was conducted using a touchscreen chamber system (Campden Instruments, Leics, UK). The operant chamber was placed inside a sound- and light-attenuating box equipped with a fan to provide ventilation and mask background noise. The touchscreen monitor (18.5 cm × 24.5 cm) was on the front of the chamber and covered by a grey plastic mask with four response windows (4 cm × 4 cm) to prevent accidental touches. The nozzle for reward delivery was located on the opposite side of the touchscreen monitor. A reward was delivered via the nozzle using a peristaltic pump through a plastic window in the wall. The trapezoidal shape of each chamber (20 cm high × 17 cm long from the screen to the food tray × 25 cm wide for the screen and 6 cm wide for the food tray) facilitated the animal’s focus on the touchscreen and reward delivery area. The top of the chamber was covered with a transparent plastic lid.

#### 4.6.2. IGT Pre-Training Session

The protocol was previously described in [58,59], with minor modifications. Access to food and water was restricted for 2 h each day at least 1 week prior to pre-training to provide sufficient motivation to perform the task. Food and water restrictions were continued until the end of the task. The body weight was maintained at 85–90% of non-restricted mice. The task started with a 5-stage pre-training phase designed to shape screen-touch behavior in mice. In stage 1, mice were habituated to the touchscreen chamber on 3 consecutive days. On day 1, mice were allowed to freely explore the chamber for 10 min without stimulus or reward presentation (strawberry milkshake). On day 2, mice were left in the chamber for 20 min. Initially, the food tray was primed with 150 μL of reward delivery and the tray light was activated. When mice collected the reward and left the tray, the tray light was deactivated. A 10 s delay was inserted before the tray light was activated, and 7 μL of the reward was then delivered. This procedure was repeated until the session ended. On day 3, the same procedure was performed for 40 min. In stage 2, mice were trained to learn the relationship between the light stimulus on the screen and the reward. When mice entered the food tray, the stimulus (a white square) was displayed randomly in one of the four windows for 30 s. If the mice touched the window, they were given 7 μL of the reward. Even if they failed to do so, they were rewarded 30 s after the trial initiation. After collection of the rewards, the next trial was started with the presentation of the stimulus image. The criterion was the completion of 30 trials within a 30 min session. In stage 3, the mice were required to touch the screen to receive the reward. The stimulus was presented in each of the four windows, and the mice were required to touch one of the four windows to receive the reward. After entry to collect the reward, the tray light was turned off, and an inter-trial interval (ITI) of 5 s was started. Then, another stimulus was displayed following the ITI period. The criterion was to collect 40 rewards within a 30 min session on 2 consecutive days. In stage 4, mice serially learned to touch one of the four windows that were randomly illuminated for different stimulus duration times (starting from 32 s, then serially reduced to 16 s, and finally 10 s) to receive the reward. Following an ITI of 5 s, a stimulus was presented in one of the four windows. Mice were required to respond within a defined period. A correct response, touching the location where the stimulus was presented, triggered the presentation of reward. An incorrect response, touching a location other than where the stimulus was presented, or making no response at all (an omission) resulted in a 5 s time out (TO) with the illumination of the house light. A premature response was recorded by touching any cue window during an ITI and resulted in a TO. Mice were required to complete the task within 100 trials or within 30 min, whichever occurred first. The criterion for success was an accuracy greater than 80% and an omission less than 20% under the conditions of an ITI of 5 s and a stimulus duration of 10 s.

#### 4.6.3. IGT Training Session

The forced-choice IGT is designed to familiarize the mice with the probabilities of receiving a reward or punishment for each window over the course of 3 consecutive days. Following an ITI of 5 s, a stimulus was presented in one of the four window panels. Mice were required to respond within a specified period (limited hold; 10 s). If the trial was rewarded (win), then the appropriate volume of reward was delivered. If the trial was punished (loss), then the image touched was flashed (0.2 s on and 0.2 s off) for the designated TO period. The reward/punishment ratio for each window was as follows: probability pattern 1 (P1), 7 μL of reward (90%) or 5 s TO (10%); probability pattern 2 (P2), 14 μL of reward (80%) or 10 s TO (20%); probability pattern 3 (P3), 21 μL of reward (50%) or 30 s TO (50%); and probability pattern 4 (P4), 28 μL of reward (40%) or 40 s TO (60%). The P2 choice yielded the optimal reward per unit of time, with the magnitude of the net gain of each choice as follows: P2 > P1 > P3 > P4 (Figure 1). In the free-choice IGT, all four windows were illuminated simultaneously at the onset of each new trial. Following an ITI of 5 s, mice were permitted to choose one of the four window panels, which was illuminated for 10 s. The reward and punishment probabilities designated for each window were identical to those of the forced-choice IGT. Depending on the window the mice selected, they received either a reward or a punishment (TO) with probabilities programmed differently. Mice completed the task either within 100 trials or within 30 min, whichever occurred first. The percentage of choices was used to measure the animals’ preferences for the different windows. To eliminate any potential location bias, the windows were allocated in a counterbalanced manner. Half of the animals were assigned windows 1 (P1), 2 (P4), 3 (P2), and 4 (P3), while the other half were assigned windows 1 (P3), 2 (P2), 3 (P4), and 4 (P1). The criterion was to continue with daily testing until a statistically stable pattern of responding (coefficient of variation value < 10) was observed over 3 consecutive sessions. Decision-making was evaluated as a percentage of the disadvantageous choices (P3 + P4)/(P1 + P2 + P3 + P4) × 100. Several other behavioral metrics including premature response, omission, and collection latency were recorded and presented (Table 1).

#### 4.6.4. IGT Test Session

After meeting the criteria in the free-choice IGT, mice underwent the 6-OHDA lesion. Two weeks later, they were tested again in the IGT (Figure 2A). In the PPX study, mice received PPX treatment for 10 days and underwent a series of tests, consisting of 5 first-half testing days (Period 1) followed by 5 second-half testing days (Period 2). For the chemogenetic manipulation of 6-OHDA-lesioned mice in the IGT, prior to the training, AAV-SYN1-KORD-P2A-mCherry was injected bilaterally into the GPe or LS (Figure 5B and Appendix A).

### 4.7. Immunohistochemistry

Immunohistochemistry was performed in accordance with a previous report [60]. Briefly, two hours after the end of the final session, mice were anesthetized with a mixture of anesthetic agents and perfused transcardially with ice-cold phosphate-buffered saline (PBS) followed by 4% paraformaldehyde in PBS. The brains were then removed and post-fixed in 4% paraformaldehyde overnight at 4 °C. The post-fixed tissues were cryoprotected in PBS containing 30% sucrose overnight and sliced into 30-μm-thick coronal sections using a sliding microtome (Leica SM2000R, Wetzlar, Germany) for immunohistochemistry.

The coronal sections were washed with PBS containing 0.3% Triton X-100, incubated at 25 °C for 2 h in 5% normal horse serum (Vectastain ABC kit, Vector Laboratories, Newark, CA, USA, #PK-6102), then incubated with a rabbit anti-TH antibody (Millipore, #AB152, diluted 1:1000) at 4 °C overnight. After washing with PBS, the sections were incubated with a biotinylated goat anti-mouse IgG secondary antibody (Vectastain ABC kit, diluted 1:1000) at room temperature for 2 h. Subsequently, the sections were incubated with PBS containing 0.3% hydrogen peroxide (Wako Pure Chemical Industries, Osaka, Japan) for 30 min to inactivate endogenous peroxidase. Sections were washed with PBS and incubated with the avidin-conjugated horseradish peroxidase complex (Vectastain ABC kit) at room temperature for 2 h. The signal was visualized using the diaminobenzidine-nickel staining method. The sections were stained using 1 mg/mL 3,30-diaminobenzidine (DAB, Sigma-Aldrich, St. Louis, MO, USA), 0.03% hydrogen peroxide, and 0.04% nickel chloride (Wako Pure Chemical Industries). These sections were mounted on slides and visualized using brightfield microscopy (BZ-9000, KEYENCE, Osaka, Japan). The density of TH-positive cells in the STR was analyzed using ImageJ 1.52k software. An average of 3 slices in each mouse was calculated and used for statistical analysis.

For c-Fos mapping with fluorescence, samples underwent a 1 h incubation in a blocking solution (5% normal goat serum) (Vector Laboratories, #S-1000-20) and incubated with a rabbit anti-c-Fos antibody (Synaptic Systems, Göttingen, Germany, #226008, diluted 1:1000) at 4 °C overnight. After washing in PBS, Alexa 594 donkey anti-rabbit IgG (Invitrogen, Carlsbad, CA, USA, #A21207, diluted 1:1000) or Alexa 488 goat anti-rabbit IgG (Invitrogen, #A11034, diluted 1:1000) was added to the sections at room temperature for 1 h. These sections were mounted on slides and visualized under a slide scanner (Axioscan 7, Zeiss, Jena, Germany). The number of c-Fos-positive cells was analyzed using NeuroInfo 2023.1.1 software. For KORD validation, the percentage of double-positive cells for mCherry and c-Fos among the total number of mCherry-positive cells was analyzed using the ImageJ software. Averages of at least three slices in each mouse were counted and used for statistical analysis.

### 4.8. Statistical Analysis

All data are expressed as the mean ± standard error of the mean (SEM). Statistical analyses were performed using the GraphPad Prism 8 software (GraphPad Software Inc., San Diego, CA, USA). One-way or two-way ANOVA tests, followed by Bonferroni’s or Dunnett’s post hoc test were used for statistical analyses with multiple group comparisons. Differences between the two groups were analyzed using a two-tailed Student’s *t*-test or Mann–Whitney U-test (parametric or nonparametric data, respectively). Detailed statistical methods are described in Appendix A.

## Figures and Tables

**Figure 1 ijms-25-08849-f001:**
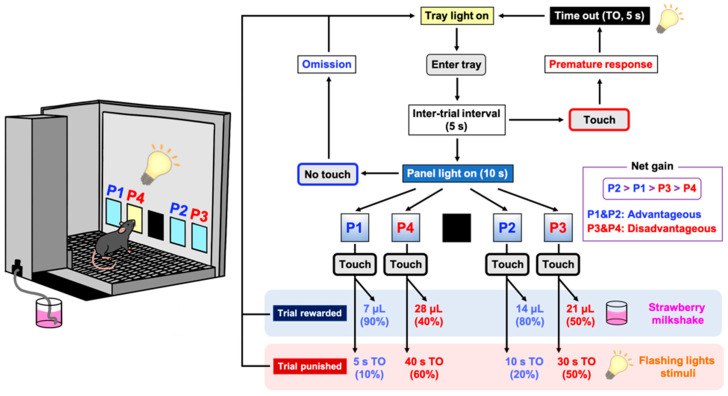
Schematic representation of a touchscreen-based Iowa Gambling Task (IGT). Each trial began with mice entering the lit food tray. After an inter-trial interval (ITI) of 5 s, all four windows (P1–P4) were illuminated simultaneously for 10 s. A premature response was recorded by mice touching any cue window during the ITI and resulted in a 5 s time out (TO). An omission was recorded when mice made no response to cue windows at all. If the trial was rewarded (win), then the appropriate volume of reward (strawberry milkshake) was delivered. If the trial was punished (loss) then the image touched flashed for the designated TO period. The reward/punishment ratio of each window was programmed differently in P1–P4. Two choices (P1, P2) resulted in the delivery of small rewards and low punishments, which led to an advantageous outcome. In contrast, the other two choices (P3, P4) resulted in the delivery of high rewards and high punishments, which led to a disadvantageous outcome. Mice completed the task either within 100 trials or within 30 min, whichever occurred first.

**Figure 2 ijms-25-08849-f002:**
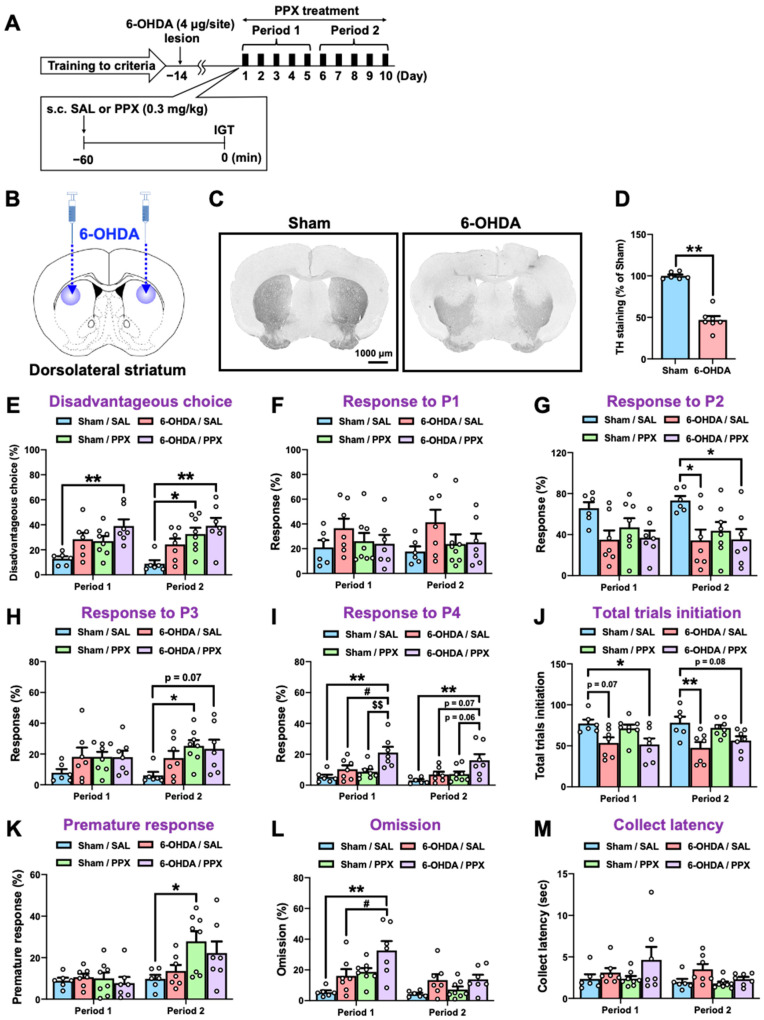
Effect of PPX treatment on decision-making in sham- and 6-OHDA-lesioned mice. (**A**) Experimental schedule. Following the fulfillment of criteria in IGT training, sham- or 6-OHDA-lesioned mice were administered with PPX (0.3 mg/kg) for 10 days, and their decision-making, motor impulsivity, attention, and motivation were assessed by the IGT over two distinct periods: the initial 5 days (Period 1) and the subsequent 5 days (Period 2). (**B**–**D**) Dopaminergic denervation in 6-OHDA-lesioned mice was evaluated. (**B**) The injection site of 6-OHDA. (**C**) Bright-field microscopic images of TH-positive cells. Scale bar = 1000 μm. (**D**) The density of TH-positive cells in the STR was analyzed. Each column represents the mean ± SEM (*n* = 6–7). ** *p* < 0.01 versus sham. (**E**–**M**) Decision-making was evaluated by the percentage of (**E**) disadvantageous choice, (**F**) response to P1, (**G**) response to P2, (**H**) response to P3, and (**I**) response to P4. The degree of session completion was shown as the number of (**J**) total trials initiation. Motor impulsivity was evaluated using (**K**) the percentage of premature response. Inattention and motivation were evaluated using (**L**) the percentage of omission or (**M**) latency to collect rewards. Each column represents the mean ± SEM (*n* = 6~8). * *p* < 0.05, ** *p* < 0.01 versus sham/SAL. ^#^
*p* < 0.05 versus 6-OHDA/SAL. ^$$^
*p* < 0.01 versus Sham/PPX. IGT, Iowa Gambling Task; 6-OHDA, 6-hydroxydopamine; SAL, saline; PPX, pramipexole; STR, striatum; TH, tyrosine hydroxylase.

**Figure 3 ijms-25-08849-f003:**
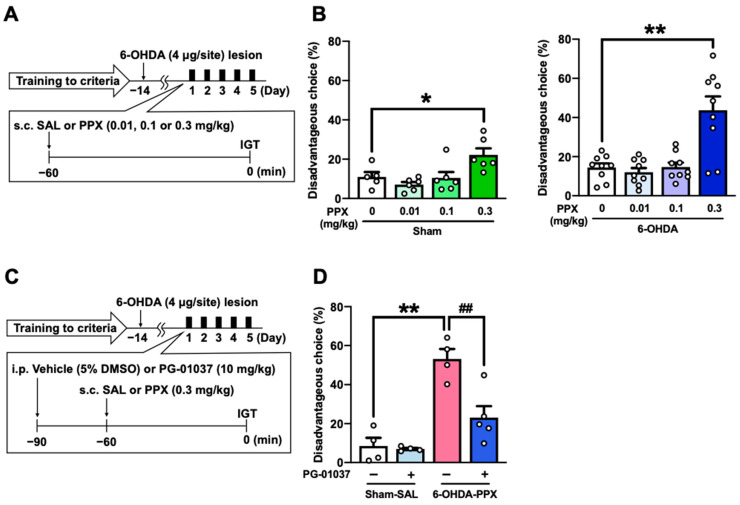
Effect of D3R antagonist on PPX-induced decision-making impairment in 6-OHDA-lesioned mice. (**A**,**B**) Dose–response relationship of PPX-induced impairment of decision-making. (**A**) Experimental schedule. After meeting the criteria of the IGT training, sham- and 6-OHDA-lesioned mice were administered PPX (0.01, 0.1 or 0.3 mg/kg) for 5 days, and their decision-making was assessed using (**B**) the percentage of disadvantageous choices in the IGT. Each column represents the mean ± SEM (*n* = 5~9). * *p* < 0.05, ** *p* < 0.01 versus PPX (0 mg/kg). (**C**,**D**) Effect of D3R antagonist on PPX-induced decision-making impairment in 6-OHDA-lesioned mice. (**C**) Experimental schedule. Sham- or 6-OHDA-lesioned mice were co-administered PG-01037 (10 mg/kg) and PPX (0.3 mg/kg) for 5 days, and their decision-making was assessed using (**D**) the percentage of disadvantageous choices in the IGT. Each column represents the mean ± SEM (*n* = 4~5). ** *p* < 0.01 versus sham-SAL/vehicle. ^##^
*p* < 0.01 versus 6-OHDA-PPX/vehicle. IGT, Iowa Gambling Task; 6-OHDA, 6-hydroxydopamine; SAL, saline; PPX, pramipexole.

**Figure 4 ijms-25-08849-f004:**
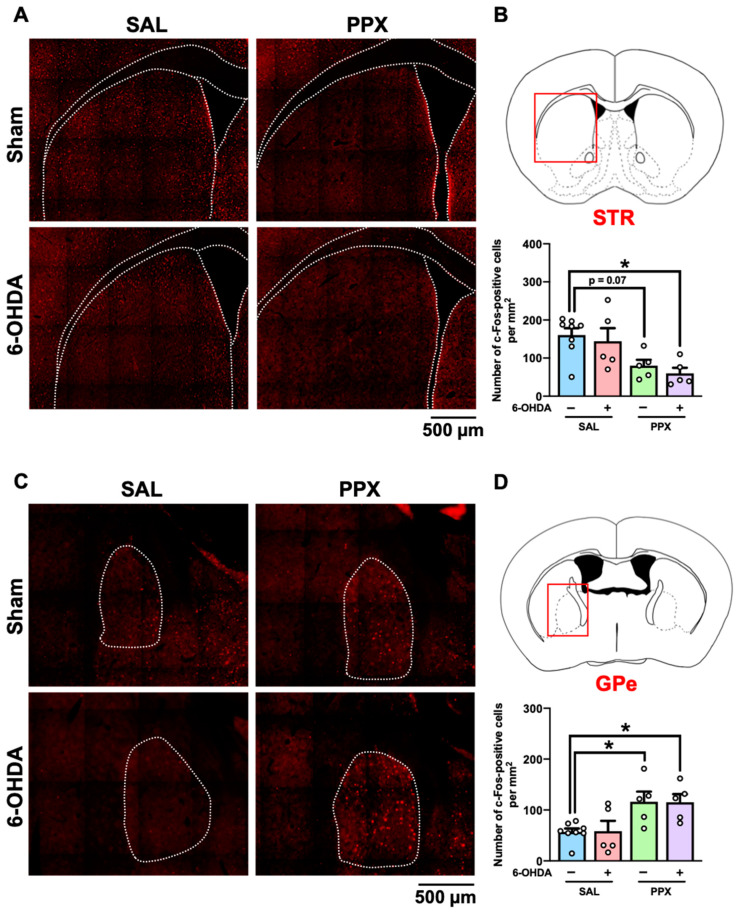
c-Fos mapping in the brain of PPX-treated sham- and 6-OHDA-lesioned mice. (**A**–**D**) Brain samples were collected 2 h after the IGT on the last day of Period 2. (**A**,**B**) c-Fos-positive cells in the STR. (**A**) Representative photographs of c-Fos-positive cells in the STR. Scale bar = 500 μm. (**B**) The red box indicates the STR. Number of c-Fos-positive cells in the STR was analyzed. Each column represents the mean ± SEM (*n* = 5~8). * *p* < 0.05 versus sham/SAL. (**C**,**D**) c-Fos-positive cells in the GPe. (**C**) Representative photographs of c-Fos-positive cells in the GPe. Scale bar = 500 μm. (**D**) The red box indicates the GPe. Number of c-Fos-positive cells in the GPe was analyzed. Each column represents the mean ± SEM (*n* = 5~8). * *p* < 0.05 versus sham/SAL. STR, striatum; GPe, external globus pallidus; 6-OHDA, 6-hydroxydopamine; SAL, saline; PPX, pramipexole.

**Figure 5 ijms-25-08849-f005:**
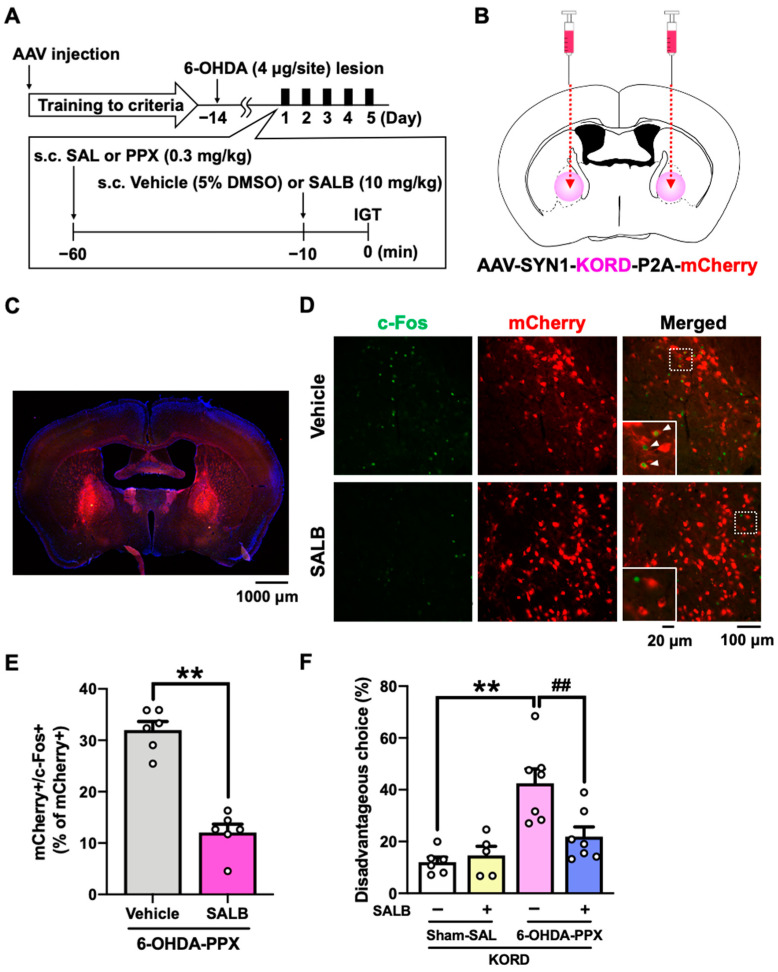
Effect of chemogenetic inhibition of the GPe on PPX-induced decision-making impairments in 6-OHDA-lesioned mice. (**A**) Experimental schedule. Prior to the IGT training, AAV-SYN1-KORD-P2A-mCherry was injected bilaterally into the GPe. After meeting the criteria in the IGT training, 6-OHDA-lesioned mice were co-administered PPX (0.3 mg/kg) and SALB (10 mg/kg) for 5 days, and their decision-making was assessed in the IGT. (**B**) The injection site of AAV-SYN1-KORD-P2A-mCherry. (**C**) Representative photograph of the expression of KORD-mCherry in the GPe. Scale bar = 1000 μm. (**D**) Representative photographs of c-Fos- and mCherry-positive cells after the vehicle or SALB (10 mg/kg) treatment in the GPe of KORD-mCherry-expressing mice treated with PPX. Scale bar = 100 μm. Each image located in the bottom left corner is an enlarged image of the dashed square. Scale bar = 20 μm. The white triangles indicate double-positive cells for mCherry and c-Fos. (**E**) The percentage of double-positive cells for mCherry and c-Fos among the total number of mCherry positive cells in the GPe of 6-OHDA-lesioned mice treated with PPX was analyzed. Each column represents the mean ± SEM (*n* = 6). ** *p* < 0.01 versus 6-OHDA-PPX/vehicle. (**F**) Decision-making was evaluated using the percentage of disadvantageous choices. Each column represents the mean ± SEM (*n* = 5~7). ** *p* < 0.01 versus sham-SAL/vehicle. ^##^ *p* < 0.01 versus 6-OHDA-PPX/vehicle. GPe, external globus pallidus; KORD, κ-opioid receptor-based DREADD; IGT, Iowa Gambling Task; 6-OHDA, 6-hydroxydopamine; SAL, saline; PPX, pramipexole; SALB, salvinorin B.

**Figure 6 ijms-25-08849-f006:**
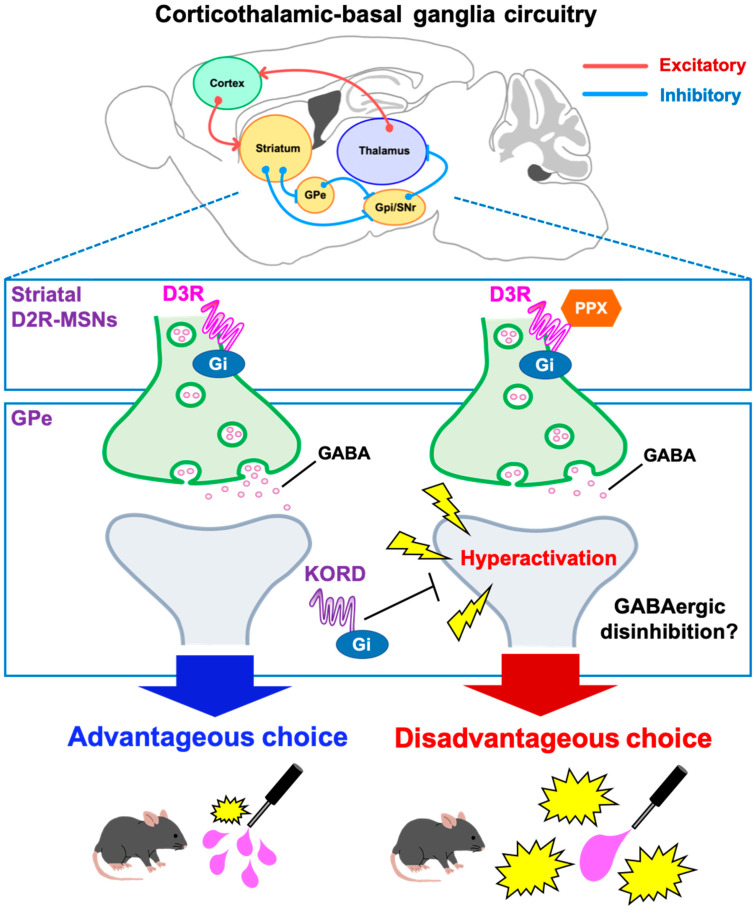
Schematic diagram of a potential mechanism for PPX-induced decision-making impairments in 6-OHDA-lesioned mice. In the corticothalamic-basal ganglia circuitry, GABArgic D2R-MSNs of the indirect pathway inhibit neuronal activity of the GPe. D2R-MSNs express the Gi-coupled D3R, which can inhibit intracellular signaling. Accordingly, the D3R-preferring agonist PPX disinhibits GABAergic D2R-MSNs, leading to hyperactivation of the GPe. Heightened activity in the GPe impairs decision-making, which was validated by the DREADD system. The activity of the D2R-MSNs-GPe circuit may be crucial for learning negative outcomes to modify future actions. D2R-MSNs, dopamine D2 receptor-expressing medium spiny neurons; GPe, external globus pallidus; GPi, internal globus pallidus; SNr, substantia nigra pars reticulata; PPX, pramipexole; D3R, dopamine D3 receptor; DREADD, designer receptors exclusively activated by designer drugs; KORD, κ-opioid receptor-based DREADD.

**Table 1 ijms-25-08849-t001:** Description of the behavioral metrics employed in the touchscreen-based IGT.

Measure & Description	Outcome
Total trials initiation: Number of trials completed per session (Max = 100)	Degree of session completion
Response (%): Response to P1, P2, P3 or P4 as a percentage of the total cue windows touched	Preference for 1 cue compared to others
Disadvantageous choice (%): Disadvantageous response options (P3 + P4)/total (P1 + P2 + P3 + P4) × 100	Risk preference
Premature response (%): Responding in any cue window during the inter-trial interval of 5 s	Motoric impulsivity
Omission (%): Failure to respond in any cue window during the light stimulus duration of 10 s	Inattention/amotivation
Collect latency: The latency to collect rewards after responding in a cue window	Motivation

## Data Availability

The datasets are available from the corresponding author upon reasonable request.

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
