# Peer review of "Pramipexole Hyperactivates the External Globus Pallidus and Impairs Decision-Making in a Mouse Model of Parkinson’s Disease"

_ijms, 2024, doi:10.3390/ijms25168849_

Round 1

Reviewer 1 Report (Previous Reviewer 2)

Comments and Suggestions for Authors

The authors have addressed all the concerns and made necessary changes in the manuscript. The revised manuscript can be accepted for publication. 

Author Response

Reviewer: 1

Comments and Suggestions for Authors

The authors have addressed all the concerns and made necessary changes in the manuscript. The revised manuscript can be accepted for publication.

Response 1: Thank you for the positive remarks.

Reviewer 2 Report (New Reviewer)

Comments and Suggestions for Authors

The manuscript presented by Kobuta and colleagues is well organized and well written, with clear methods and results section. The topic of improving pharmacological intervention in PD to reduce behavioral addictions is rising interest, making this study well-deserving publication.

According to my experience two discussion points are currently missing/insufficient and they should be included in the introduction and/or discussion of the manuscript before final publication:

1)  Impairment of higher cortical regions in PD are currently proven, also in mouse and rat models. Undoubtedly, this is involved in the study framework, and it should be cited (for example doi: 10.1093/brain/awz243; doi: 10.3390/cells11172628 and doi: 10.1016/j.neuroscience.2009.04.069), since it is finally affecting the pharmacological effect. 

2) Behavioral addictions induced by dopaminergic agonism are usually referred to the other main dopaminergic source (the VTA and the mesolimbic circuit), which underlies reward/motivation/positive reinforcement, and it is not specifically degenerated in PD. This should be clearly mentioned and discussed, beyond the "decision making" process already stated. Lots of studies in mice/rats can be cited according to the author’s preference.

Author Response

Reviewer: 2

Comments and Suggestions for Authors

The manuscript presented by Kobuta and colleagues is well organized and well written, with clear methods and results section. The topic of improving pharmacological intervention in PD to reduce behavioral addictions is rising interest, making this study well-deserving publication.

According to my experience two discussion points are currently missing/insufficient and they should be included in the introduction and/or discussion of the manuscript before final publication:

Comment 2-1: Impairment of higher cortical regions in PD are currently proven, also in mouse and rat models. Undoubtedly, this is involved in the study framework, and it should be cited (for example doi: 10.1093/brain/awz243; doi: 10.3390/cells11172628 and doi: 10.1016/j.neuroscience.2009.04.069), since it is finally affecting the pharmacological effect.

Response 2-1: We are grateful for your valuable suggestion. In accordance with the comment from reviewer 2, we have discussed the following: “Impairment of higher cortical regions involved in decision-making has been demonstrated in rodent models of PD [Wang et al., Neuroscience 2009, 162, (4), 1091-100; Li et al., Brain 2019, 142, (10), 3099-3115; Palmas et al., Cells 2022, 11, (17)]. This may support, at least in part, our working model (Figure 6) and contribute to the potentiating effect of PPX in 6-OHDA-lesioned mice. Another hypothesis is that D3R expression and activity could be altered in 6-OHDA-lesioned mice.’’ (Lines 288-292).

Comment 2-2:

Behavioral addictions induced by dopaminergic agonism are usually referred to the other main dopaminergic source (the VTA and the mesolimbic circuit), which underlies reward/motivation/positive reinforcement, and it is not specifically degenerated in PD. This should be clearly mentioned and discussed, beyond the "decision making" process already stated. Lots of studies in mice/rats can be cited according to the author’s preference.

Response 2-2: In accordance with the comment from reviewer 2, we have added a paragraph discussing behavioral addictions as follows: “Behavioral addictions induced by dopamine agonism are referred to the other major dopamine sources, including the mesolimbic system from the ventral tegmental area (VTA) to the nucleus accumbens (NAc), which is not specifically degenerated in PD. In this system, the NAc D1R-MSN is responsible for regulating reward, motivation, and positive reinforcement [Zhang et al., Int J Mol Sci 2022, 23, (19); Tsuboi et al., ell Rep 2022, 40, (10), 111309]. Our c-Fos mapping did not reveal any alteration in the VTA and NAc of PPX-treated mice (Table S1). Accordingly, the mechanism by which PPX induced decision-making impairments may differ from that of behavioral addictions. However, a different interpretation may be provided by real-time observation of individual neuronal activity in the mesolimbic system during the IGT by in vivo recording (e.g. calcium imaging). This should be investigated in future studies.” (Lines 253-262).

This manuscript is a resubmission of an earlier submission. The following is a list of the peer review reports and author responses from that submission.

Round 1

Reviewer 1 Report

Comments and Suggestions for Authors

The purpose of the study was to examine the influence of the dopamine D3 receptor (D3R)-preferring agonist 13 pramipexole (PPX) on decision-making. This was accomplished in a mouse PD model. PPX was given to the mice after the induction of PD and the mice performed a version of the Iowa Gambling Task developed for mice to assess decision making. The main findings were that PPX increased disadvantageous choices characterized by high-risk/high-reward in the touchscreen-based Iowa Gambling Task and PPX enhanced the number of c-Fos-positive cells in the GPE. Overall, the authors concluded that hyperactivation of GPe neurons in the indirect pathway impairs decision-making in PD model mice, which may have implications for gambling in PD patients.

Overall, this was a very extensive and work intensive study and itseemed to be conducted carefully, was easy to understand, had a solid design, and was very well-written with few if any grammatical or typographical errors. I think the study adds to the literature on the topics of gambling and risk taking behavior in PD due to medications. The study should be of interest to readers of IJMS and researchers is several related fields. The mouse variation of the Iowa Gambling Task is novel. The findings have relevance to pathological gambling and risk taking in PD patients. Finally, the figures were very well done. 

I only have a few minor comments of various types that the authors should consider and a few minor corrections to suggest. 

  1. Is it the journals format to have the Results section after the Introduction or should the Methods go there?
  2. Lines 131 to 150 should these statistical results be put in a table to make them easier to read? Maybe highlighting in bold the significant results also once it is in a table. It is very hard to read and follow when put in a figure legend like this. The same comment could be made for other figures but not to the same extent.
  3. The study had very few typos or other types of mistakes. There do appear to be a few in the bibliography where a few references has the journal article title with capitalized first letters of all words, but most do not. For instance, references 14. 23, and 29 (as some examples, more exist) have a different format than most of the other references for the article titles.

Reviewer 2 Report

Comments and Suggestions for Authors

In the manuscript “Pramipexole hyperactivates the external globus pallidus and impairs decision-making in a mouse model of Parkinson’s disease” by Kubota et al., the authors explored the effect of Pramipexole (PPX) – a dopamine D3 receptor (D3R)-preferring agonist on decision-making ability using 6-OHDA-induced mice as a model of Parkinson’s Disease (PD). They identified impaired decision-making 0f PD mice upon treatment with PPX, as identified by Iowa Gambling Task. PPX treatment also increased the number of c-Fos positive cells in the external globus pallidus (GPe). Finally, the authors demonstrated that chemogenetic inhibition of the GPe restored impaired decision-making caused by PPX treatment. Overall, I enjoyed reading the manuscript. Majority of the experiments were performed with care and results were interpreted well except a few cases. However, in some experiments, proper control is missing which makes it difficult to interpret the results. Specific comments are summarized below.

1.        In some figures, statistical tests were performed poorly. For example, in Figure 2E, difference between Sham/SAL and 6-OHDA/PPX were shown to be statistically significant. I do not understand what the authors wanted to conclude from this statistical test! Sham/ SAL should be compared with Sham/PPX and/or 6-OHDA/ SAL (not 6-OHDA/PPX). Similarly, 6-OHDA/ SAL should be compared with Sham/ SAL and/or 6-OHDA/PPX. All statistical tests need to be revisited and authors are advised to consult with a Biostatistician.  

2.        What was the age of the mice when the experiments were performed? This is a critical information but missing in the manuscript.

3.        Figure 2 (E-I, K, L). Is the difference between Sham/SAL and 6-OHDA/SAL in different panels significant? This is extremely important in order to understand how the 6-OHDA affects decision making (or not) in the absence of PPX.

4.        Figure 3B. Why are the control (Sham) mice missing in this experiment? I am curious to know if the same doses of PPX alter decision-making in control mice.

5.        Figure 3D. Where is the Sham/PPX set? That should be the control for 6-OHDA/PPX set.

6.        Figure 2 legend is too lengthy and hard for readers to understand the values and statistics. Can the authors make a table with all the numbers and statistical values?

Reviewer 3 Report

Comments and Suggestions for Authors

Overview of the manuscript

The manuscript focuses on the investigation of D2/D3 receptors antagonists as responsible for inducing impairments in decision-making including pathological gambling. D2/D3 antagonists are commonly drugs in therapeutic protocols for Parkinson disease (PD), and pathologic gambling is observed as an adverse phenomenon in patients. The authors perform an experimental animal PD model by injecting the toxin 6-hydroxydopamine into the dorsolateral striatum bilaterally into mice. Subsequent treatment with the D3 receptor-preferring agonist PPX, increased disadvantageous choices characterized by high-risk/high-reward in the touchscreen-based Iowa Gambling Task. This effect was blocked by treatment with the selective D3R antagonist PG-01037. In mice treated with PPX, the number of c-Fos-positive cells was increased in the external globus pallidus (GPe), indicating dysregulation of the indirect pathway in the corticothalamic-basal ganglia circuitry. The authors conclude that these findings demonstrate that hyperactivation of GPe neurons in the indirect pathway impairs decision-making in PD model mice. The results provide a candidate mechanism and therapeutic target for adverse events observed during D2/D3 receptor pharmacotherapy in PD patients.

GENERAL  COMMENT

The work is very interesting, the experimental plan is rigorous in the use of methodologies and statistical analysis. The several experimental steps are adequately described and documented.

Following I suggest some improvement indications for better readability of the work.

 Specific coMments

Results

Take care to make the colour legend legible in the several histograms presented in the work.

Fig. 6: correct “Strital”

Materials and Methods

Pag. 12, line 373 – 375: the importance and use of the described experimental procedure should be presented in a more extensive way.

Pag. 14, line 482 – 497: this paragraph details the immunoistochemical procedure against TH. That is presented in Figure 2, I suppose. But you state the use of fluorescence microscope. This is not possible and is not compatible with the images presented. Correct the paragraph or explain better.